# Biomechanical, Microstructural and Material Properties of Tendon and Bone in the Young Oim Mice Model of Osteogenesis Imperfecta

**DOI:** 10.3390/ijms23179928

**Published:** 2022-09-01

**Authors:** Antoine Chretien, Malory Couchot, Guillaume Mabilleau, Catherine Behets

**Affiliations:** 1Pole of Morphology, Institute of Experimental and Clinical Research (IREC), UCLouvain, 1200 Brussels, Belgium; 2Univ Angers, Nantes Université, Oniris, Inserm, RMeS, REGOS, SFR ICAT, F-49000 Angers, France; 3Department of Cell and Tissue Pathology, CHU Angers, F-49933 Angers, France

**Keywords:** osteogenesis imperfecta, oim, bone, tendon, biomechanics, Raman

## Abstract

Osteogenesis imperfecta (OI) is a genetic disorder of connective tissue characterized by low bone mass and spontaneous fractures, as well as extra-skeletal manifestations, such as dental abnormalities, blue sclera, hearing loss and joint hypermobility. Tendon ruptures have been reported in OI patients. Here, we characterized the biomechanical, structural and tissue material properties of bone and tendon in 5-week-old female osteogenesis imperfecta mice (oim), a validated model of severe type III OI, and compared these data with age- and sex-matched WT littermates. Oim tendons were less rigid and less resistant than those of WT mice. They also presented a significantly higher rate of pentosidine, without significant modification of enzymatic crosslinking. The oim bones were less resistant and avulsion fractures were evident at high tendinous stress areas. Alterations of trabecular and cortical bone microarchitectures were noticed in young female oim. Bone tissue material properties were also modified, with a less mature and more mineralized matrix in association with lower collagen maturity. Our data suggest that the tendon-to-bone unit is affected in young oim mice, which could explain tendon ruptures and bone fragility observed in OI patients.

## 1. Introduction

Osteogenesis imperfecta (OI) is a disorder of connective tissue generally associated with dominantly inherited pathogenic variants in *COL1A1* or *COL1A2* genes. Consequently, defects in collagen molecular structure, and possibly mineral alterations, lead to decreased bone strength, resulting in low-trauma fractures, skeletal deformities and growth deficiency [1]. The clinical classification of OI, established by Sillence in 1979, includes four groups: type I, a mild, non-deforming phenotype; type II, associated with perinatal lethality; type III, a severe, non-lethal, progressive deforming phenotype; and type IV, with moderate severity [2]. This classification was later expanded according to genetic causes [3].

Aside from skeletal alterations and symptoms, teeth, which are also mineralized tissues rich in collagen type I, are commonly implicated in OI [4]. They exhibit discoloration and translucency and can be worn off prematurely. The roots are short and constricted and the dentine is hypertrophic [5]. These changes are characteristic of dentinogenesis imperfecta and can be observed either in the context of OI or as isolated. Moreover, patients may also present hearing loss, blue sclera, joint hypermobility, muscle weakness or cardiovascular abnormalities. All these manifestations are listed in Appendix A, drawing on previous work [5,6]. By extension, all tissues with a high collagen I content, such as the tendons, can be affected. Tendon ruptures have been described in patients with OI [7,8,9,10]. They were sometimes bilateral and occurred at the site of bone insertion; namely, the enthesis, which distributes the mechanical forces across the whole tendon–bone interface [11].

Bone and tendon are made up of dense, regular connective tissue deposited by specialized cells, osteoblasts and tenocytes, with organic extracellular matrices composed mainly of collagen. While type I is the major form of collagen in bone and tendon, other types are also present, including type III. Bone and tendon organic extracellular matrices also contain non-collagenous proteins, including proteoglycans and glycoproteins, whose functions are to promote cell attachment and control collagen fibrillogenesis and hydration of matrices. Among all the glycoproteins found in tendon, tenomodulin is known as a marker of tendon maturation and plays a role in proliferation [12].

However, the main difference between the two tissues is the mineralization of bone extracellular matrix by tablets of poorly crystalline substituted hydroxyapatite.

Bone and tendon mechanical properties are highly dependent on collagen structure and organisation and, for bone only, on the structure of the mineral tablets and the degree of mineralization. Some elements play a cardinal role; namely secondary conformation (triple helix and beta sheet), hydroxyproline or bound-water content, porosity and cross linking, whether enzymatic (including mature pyridinoline and immature dehydrodihydroxylysinonorleucine) or non-enzymatic, such as advanced glycation end-products (pentosidine) [13].

In a preliminary histological study, we highlighted a different proportion of thick collagen fibres, as well as fibrocartilage islets, in tendons of oim mice [14]. The osteogenesis imperfecta mouse (oim) has a genetic mutation in type I collagen that results in the exclusive production of α1(I) collagen. The homozygotes are a model of human type III osteogenesis imperfecta. Modification of the collagen structure, represented by the presence of a homotrimer of α1(I) collagen, likely underlies reduced torsional strength of the whole bone, which has been observed to decrease by about 40% in the mutants [15]. The homozygous animals are smaller in body mass and length than heterozygous and normal littermates [16], and morphometric examination of the femurs has revealed alteration of bone geometry for the oim mice [15]. Alteration of the quality of bone material has also been described in older animals and in young animals but irrespective of gender [17,18]. More recently, we showed that weight, snout–sacrum length and bone mineral content were significantly smaller for 5-week-old male oim than females ones [19]. At 14 weeks, these differences disappeared and male oim showed more numerous long bone fractures than females. However, no study has so far investigated the tendon and bone properties at the biomechanical, structural and tissue material levels in the same animals, especially at a young age.

The main objective of the present study was to characterize the biomechanical, structural and tissue material properties in 5-week-old female oim and to compare these data with age- and sex-matched WT littermates

## 2. Results

### 2.1. Tendon-to-Bone Unit Is Altered in Osteogenesis Imperfecta

As suggested by the weight difference between oim and WT mice (13.5 ± 0.6 g and 19.3 ± 0.6 g, respectively, *p* < 0.0001), CT imaging highlighted frailer but also less radiopaque bones in oim than in WT mice (Figure 1A). Fractures were visible only in oim; in particular, avulsion fractures in high tendinous stress areas, such as the calcaneus. There was, on average, more than one calcaneal fracture per mouse (1.33 ± 0.71), and these gave rise to superior concavity deformations of the bone.

As expected, bone mineral density measured with pQCT in the ulna epiphysis was significantly lower in oim than WT mice (−22.3%) (Figure 1B).

Histological analysis at the elbow highlighted very thin cortical bone, as well as thin and rare bone trabeculae, in oim ulna epiphyses (Figure 1C). The bone volume/total volume (BV/TV) ratio in the olecranon epiphysis was significantly lower in oim (−39.6%) than WT mice (Figure 1E).

Moreover, there seemed to be a relationship between the triceps brachii tendon thickness and the BV/TV in the proximal epiphysis (R^2^ = 0.21, *p* = 0.07). This suggests a veritable osteotendinous unit, likely with an adaptation of tendon development according to mechanical constraints.

### 2.2. Oim Tendons Are Less Rigid and Less Resistant

The mass (0.54 ± 0.04 mg) and the length (4.6 ± 0.2 mm) of oim flexor digitorum longus tendons were significantly lower than WT ones (1.45 ± 0.08 mg, *p* < 0.01, and 5.5 ± 0.5 mm, *p* < 0.01, respectively). As illustrated by individual stress–strain and stress evolution curves, oim tendons were less rigid and less resistant than WT ones (Figure 2A,B). On the whole, oim tendons showed significantly lower ultimate stress than WT ones (−39%) whereas their ultimate strain was higher (+27%, *p* = 0.17) (Figure 2C,D). No significant difference in toughness was evidenced between the two groups of animals, even if the data were close to the limit of significance (Figure 2E; −33%, *p* = 0.14). When the tendon was deformed to 15% of its initial length, the stress values were lower for oim tendons, indicating a lower rigidity (Figure 2B). Indeed, the elastic modulus was significantly lower in oim tendons (−60%, Figure 2F).

### 2.3. Oim Bone Strength Is Reduced

The load–deformation curves show that the oim bones had a lower deformation and fractured with a lower load than the WT bones (Figure 3A). The ultimate load (Figure 3B), the stiffness (Figure 3C), the yield load (Figure 3D) and the post-yield displacement (Figure 3E) were significantly lower, by −64%, −58%, −56% and −71%, respectively, than WT ones (*p* = 0.0002; *p* = 0.0003; *p* = 0.0021; *p* = 0.0001, respectively). As all parameters were decreased, the work to fracture (Figure 3F) was also significantly lower, by −84.4%, than in WT mice (*p* = 0.0001).

### 2.4. Bone Microarchitecture Is Altered in Oim

Assessment of bone microstructure was performed after three-point bending. Unfortunately, three oim bones broke close to the distal metaphysis and hampered the microCT analysis. Figure 4A presents a three-dimensional model of the femur metaphysis and diaphysis. As compared to WT littermate bones, oim bones appeared smaller, with lower trabecular bone mass and thinner cortices. Microarchitectural analysis (Figure 4B–D) revealed that, as compared to WT animals, oim had significant reductions in trabecular bone mass (−56%, *p* < 0.0001), trabecular number (−51%, *p* = 0.0001) and trabecular thickness (−10%, *p* = 0.008). At the cortical level (Figure 4E–I), significant reductions in Ct.Th (−21%, *p* = 0.022), Tt.Ar (−38%, *p* < 0.0001), Ma.Ar (−55%, *p* = 0.0002) and cross-sectional moment of inertia in the medio-lateral axis (−72%, *p* = 0.0003) were observed. On the other hand, cortical porosity was significantly increased by 63% (*p* = 0.004) in oim as compared with WT controls.

### 2.5. Tendon Tissue Material Properties Are Altered in Oim Mice

Tendon tissue material properties were investigated in the triceps brachii tendon (Table 1). We observed significant augmentations in non-enzymatic collagen crosslinking (+20%, *p* < 0.01) and β-sheet content (+43%, *p* < 0.05) and a significant reduction in collagen NH stretch-bound water (−37%, *p* < 0.05) in oim as compared with WT animals. None of the other studied tendon material parameters were significantly different between the two groups of animals. The expression of collagen III in oim tendons was significantly lower than in WT mice, but other components and regulator did not seem to be impacted. Indeed, expression of decorin and tenomodulin looked identical in both groups (Figure 5).

### 2.6. Bone Tissue Material Properties Are Altered in Oim Mice

We performed the analysis of bone tissue material properties at the cortical bone (Figure 6). In order to account for tissue age, analysis was undertaken between double calcein labels, representative of a tissue age of 2–10 days. Figure 6A illustrates the mean spectra for WT mice and oim. Figure 6B,C represent the subpeaks, determined after the second derivative and curve fitting, of the v1, v3 PO4 and amide I bands. Analysis of the mineral bone matrix (Figure 6D) showed significant augmentations in phosphate/amide ratio (+13%, *p* = 0.013) and HPO4 content (+29%, *p* = 0.023) and a significant reduction in the mineral crystallinity (−4.6%, *p* = 0.009) as compared with WT animals. None of the other mineral parameters were significantly different between the two groups of animals. At the organic matrix level (Figure 6E), we found a significant reduction in collagen maturity (−23%, *p* = 0.041). None of the other organic parameters showed a significant difference between the two groups of animals.

## 3. Discussion

Our study provides an in-depth investigation of bone and tendon biomechanical, structural and compositional properties in young oim. The oim tendons were slighter, less rigid and less resistant than WT ones. We also found differences in the collagen matrix, since oim presented significantly higher rates of pentosidine and beta sheets. The rates of enzymatic crosslinking were not significantly different but oim tendons were more porous and had lower bound water than WT tendons. In bone tissue, we found a significant reduction in biomechanical properties associated, with alterations of bone microstructure (trabecular and cortical bones) and bone tissue material properties, especially collagen maturity, the phosphate/amide ratio, mineral crystallinity and HPO_4_ content.

As expected, and as described in another study [19], OI-related osteopenia and lower bone density were visible in the epiphysis. It is plausible that severe bone deformations observed in the area of the tendon insertion led to motor impairment. In a recent study also interested in the impact of the disease on the tendons [20], Crtap mice (a different model of OI) presented motor impairment and thinner tendons but an increase in collagen crosslinks and a dysregulation in inflammatory signalling. However, it is worth noting that, according to the newest OI classification, this model is in fact type VII and the effects at the tissue level are probably not exactly similar.

The histological results suggested that the oim tendons were slighter than the WT ones. However, due to the mutation, oim have a short stature phenotype and the difference in tendon size could be only explained by this smaller size of the mice.

The tensile tests gave very interesting information on the mechanical properties of oim tendons. Tendon ruptures described in the literature do not seem to be due to a lack of flexibility, since the oim tendons looked more stretchable than WT ones. Based on our mechanical data, we hypothesized that collagen reticulation would be lower in oim than WT mice. Surprisingly, we did not observe differences in the rates of enzymatic cross-links between groups. In the literature, some data show that pyridinoline may be identical or higher in oim [21]. For the non-enzymatic reticulation, the rate of pentosidine was significantly higher in oim tendons, as described in oim bones [22]. Interestingly, a previous study showed lower viscoelasticity in the jawbone associated with high levels of pentosidine in the bone matrix [23]. In our young oim, this high ratio could be explained by gestational diabetes, which is known to increase levels of advanced glycation end-products, but further investigation is required to delineate the exact mechanism that leads to higher accumulation of advanced glycation end-products in oim. This could also be relative to a lower rate of tendon remodelling. However, investigating tendon remodelling is challenging due to the low amount of matrix replaced over time.

Previously, reductions in the mean crystal thickness and degree of alignment of mineral crystals, as well as an elevated degree of mineralization, were reported in oim [24]. Our results are in agreement with the higher degree of mineralization, which was mirrored, in the present study, by an increase in the phosphate/amide I ratio. However, we found a significant reduction in mineral crystallinity, which was also observed in older animals by Camacho et al. [17], and higher HPO4 content, suggesting that the maturation rate of hydroxyapatite was lower in oim. This was also in agreement with the lower crystal thickness. Interestingly, and in opposition to what was observed in the tendon, we found a significant reduction in collagen maturity in bone but no modification of collagen secondary structure. This is intriguing and an explanation might reside in the fact that the bone matrix, although similar in its organic composition to tendon, is mineralized by hydroxyapatite tablets. Further investigations are required to understand this discrepancy and to investigate whether such features are also observed in humans.

The homotrimeric collagen of oim is known to form larger kinks and to freely rotate with a much larger angle than heterotrimers. In bone, it has been shown that collagen packing is affected, since there is a larger lateral spacing in fibrils and a larger gap/overlap ratio [25]. However, in the present study, these structural modifications did not seem to change the secondary structure of mineralized collagen. The amino acids glycine and proline are very abundant in collagen sequences and are known to have a low propensity to form beta sheets. Nevertheless, our Raman spectroscopy analysis showed a difference in collagen secondary conformation, since there was a higher ratio of beta sheets in oim than in WT tendons. Finally, we observed a higher rate of porosity in oim tendons; i.e., space that was filled with free water during the lifetimes of the oim.

Rates of NH groups in collagen besides bounded water were significantly higher in oim than in WT triceps brachii tendons. The ratio 3325/2949 is a spectroscopic biomarker of collagen bound-water correlated with mechanical resistance [26], which is consistent with our tensile tests. However, the other rates of bound water were also linked with mechanical properties, but we did not observe significant differences between groups.

The biomolecular analysis showed that the non-collagenic matrices of tendons did not appear to be impacted since the amounts of tenomodulin and decorin were identical. It could be hypothesized that the decrease in mechanical properties in the tendon was almost exclusively explained by collagen structure damage. Unfortunately, it was not possible to quantify the elastin with Miller staining since there was no marking on the tendons. We observed a lower expression of collagen III in oim than in WT tendons. As this type is known to be present in cases of cicatrization and lesions, this led us to consider that the repair process occurs differently in oim.

Current treatments for OI aim to increase bone strength, prevent fracture and improve quality of life. They utilize a multidisciplinary approach include drug treatment, orthopedic surgery, physiotherapy and preventive dental care [27]. Intravenous injections of bisphosphonates are the most widely used intervention to reduce bone resorption, but their effects are still debated [28]. Novel therapies are under evaluation, including combinations of antiresorptive and anabolic drugs or stem cells therapy, which show potential positive effects [29,30]. It would be interesting to know if the treatments used also have an indirect effect on tendons. Indeed, with stronger bone, we could expect greater use through movement and, therefore, a possible improvement in tendon properties.

To conclude, our data suggest that the tendon-to-bone unit is affected in young oim, which could explain the tendon ruptures and bone fragility observed in OI patients. It would be interesting to evaluate the effects of age and treatments on this complex and whether such alterations are found in other models of OI with grading severity. Further studies conducted in humans are definitely required to evaluate whether such modifications of bone and tendon material properties are also present in human patients.

## 4. Material and Methods

### 4.1. Animal Procedures

Heterozygous mice (strain B6C3Fe a/a-Col1a2^oim^/J, JAX stock #001815) were obtained from the Jackson laboratory (Charles River Laboratories, 69592 L’Arbresle, France). Heterozygous mice were crossed, and 5-week-old female homozygous oim (*n* = 10) and wild-type littermates (*n* = 10) were used for this study. The genotype was monitored with PCR to amplify the purified genomic DNA from cut tail samples, using the following primers: forward—(5′-3′) GGCTTTCCTAGACCCCGATGCTTAG; reverse for WT—(5′-3′) GTCTTGCCCCATTCATTTGTC; and reverse primer for OI—GTCTTGCCCCATTCATTTGTT. All procedures were undertaken in accordance with the Belgian federal law and protocols for the care of animals, and handling and care of mice were approved by the ethics committee for animal research of the Université Catholique de Louvain (2020/UCL/MD/02). The mice were housed at 24 °C with a 12–12 h light–dark cycle and received standard pellet chow (Safe Diets, Premium Scientific, A03) and water ad libitum. Mice received subcutaneous calcein (30 mg/kg) 10 days and 2 days before euthanasia. On the day of euthanasia, at 5 weeks, mice were weighed and scanned with an in vivo computed tomograph (Nano SPECT/ct, Mediso) under sevoflurane overdose inhalation. Elbows, bone samples, tail and flexor digitorum longus tendons were dissected within 10 min after death.

### 4.2. Investigation of Tendon-to-Bone Characteristics at the Elbow

The ulna epiphysis was scanned with a pQCT Research SA+ (Stratec, Birkenfeld, Germany). Eight transverse slices were obtained with thickness of 150 µm; the voxel size was 0.07 mm and the threshold was 280 mg/cm^3^. Bone mineral density (BMD) was assessed in three transverse slices of each bone with the XCT540 software of the pQCT. After dissection, the elbow was fixed in 4% formalin, decalcified, embedded in paraffin and sliced into 5 µm thick sections, which were stained with Sirius Red and then observed under an optical microscope. The bone volume/total volume (BV/TV) ratio of the olecranon epiphysis and the triceps brachii tendon thickness were measured in the same sections with ImageJ software (version 1.52A).

### 4.3. Mechanical Test

Flexor digitorum longus tendons were used for the tensile test with a mechanical device developed in a laboratory of our institution [31] and equipped with PicoScope6 software. One end of the tendon was tied to an isometric force transducer and the other to an electromagnetic motor and length transducer. During the tests, tendons were kept hydrated in PBS at 20 °C. Mass and length were measured to determine the cross-sectional area (CSA). Tendons underwent a tensile test composed of three steps represented by loading in traction, a holding period of 20 s and unloading. After this test, additional traction of 0.2 mm was applied and repeated until fracture with a speed of 0.3 mm per second. The strain (deformation) was expressed as a percentage of the initial length. Stress–strain curves were plotted and several parameters were computed, including the slope of the linear part that corresponds to the elastic modulus, the ultimate point corresponding to the moment when the tendon is ruptured and the area under the curve representing the tendon toughness.

Three-point bending was performed on the right femur as previously reported [32]. Bones were tested in the antero-posterior axis with the posterior surface facing upward and centred on the support with a span length of 6 mm. The pressing force was applied vertically in the middle of the bone. The strength of each bone was measured with a loading speed of 1 mm/min until failure. A 500 N load cell of an Instron 5942 device (Instron, Elancourt, France) was applied and the load–displacement curve was recorded at a 100 Hz rate with Bluedhill 3 software (Instron). Ultimate load, yield load, stiffness, post-yield displacement and work-to-fracture were computed from the load–displacement curve.

### 4.4. High Resolution X-ray Microcomputed Tomography

MicroCT analyses were performed on the right femur after three-point bending with a Skyscan 1272 microtomograph (Bruker-Skyscan, Kontich, Belgium) operated at 70 kV and 140 μA, with 1000 ms integration time. The isotropic pixel size was fixed at 4 μm and the rotation step at 0.25°, and exposure was performed with a 0.5 mm aluminium filter. The trabecular volume of interest was located 0.5 mm above the growth plate at the distal end and extended 2 mm up. The cortical volume of interest was located 3 mm above the distal growth plate and extended by 0.5 mm. All histomorphometry parameters were measured according to guidelines and nomenclature proposed by the American Society for Bone and Mineral Research [33].

### 4.5. Tissue Material Properties Investigation

After dissection, the elbow was fixed in 4% formalin, decalcified and embedded in paraffin. Paraffin blocks were trimmed and the triceps brachii tendon composition was analysed by confocal Raman microspectroscopy using a Renishaw InVia Qontor microscope equipped with a 785 nm laser diode (Renishaw Plc, Wotton-under-Edge, UK). Spectral acquisitions were made with a laser power set at 10 mW using a 1200 lines/mm holographic grating and a 20× objective (numerical aperture: 0.4). Prior to acquisition, the Raman system was calibrated with a silicium standard at the 520 nm peak. In order to estimate tendon tissue material properties, three spectra were recorded for each tendon in the range from 800 to 1800 cm^−1^, with an integration time of 5 s and 15 accumulations. Enzymatic collagen crosslinking (intensity: ratio: 1670/1690 cm^−1^), non-enzymatic collagen crosslinking (intensity ratio 1345/920 cm^−1^), hydroxyproline content (intensity ratio: 872/920 cm^−1^), nanoporosity (intensity ratio: 1296/920 cm^−1^), glycosaminoglycan content (intensity ratio: 1380/920 cm^−1^) and β-sheet content (area ratio: 1610/total amide I) were computed. In order to estimate organic matrix-related water, we also recorded three spectra per tendon in the spectral range from 2700 to 3800 cm^−1^, following Unal et al. [34]. Briefly, integration times of 10 s and six accumulations were used. All tendon spectra were scaled to obtain the same intensity at the CH stretch peak of collagen at ~2940 cm^−1^. Collagen-bound water (intensity ratio: 3243/2940), collagen NH stretch (intensity ratio: 3330/2940), and collagen OH stretch (intensity ratio: 3457/2940) were computed.

Right femurs were embedded undecalcified in pMMA after dehydration and infiltration, as previously reported [35]. One micrometre-thick sections of the midshaft femur were cut with an ultramicrotome (Leica EM UC7, Leica microsystems, Nanterre, France) and deposited on BaF2 windows. Spectral analysis was performed with a Bruker Vertex 70 infrared spectrometer coupled to a Bruker Hyperion 3000 microscope equipped with a single-element mercury-cadmium-telluride infrared detector (Bruker France SAS, Champs sur Marne, France). Spectra were recorded between double calcein labelling. Each spectrum was corrected for Mie scattering. The contribution of pMMA at ~1730 cm^−1^ was computed and a nanoporosity ratio (area ratio 1730 cm^−1^/v1,v3 PO4) was calculated. After pMMA subtraction, the spectrum was vector-normalized for the v1, v3 phosphate band and analysed with a routine script in Matlab R2021b (The Mathworks, Natick, MA, USA), as previously reported [35]. Phosphate/amide I ratio (area of v1,v3 phosphate/area amide I), mineral crystallinity (1/full width at half-maximum of the v1 PO_4_ peak at 960 cm^−1^), crystal size index (area ratio: ~1076 cm^−1^/~1058 cm^−1^), acid phosphate content (area ratio: ~1127 cm^−1^/~1096 cm^−1^), carbonate/phosphate ratio (intensity of v2 carbonate located at 850–900 cm^−1^/v1,v3 PO4 band) and collagen maturity (area ratio: ~1660 cm^−1^/1690 cm^−1^) were computed. The secondary structure of collagen matrices was also computed following Belbachir et al. [36].

### 4.6. Western Blot

The tail tendons were homogenized in radioimmunoprecipitation assay buffer containing a protease inhibitor cocktail and PhoSTOP (Roche, Basel, Switzerland). Protein concentration was determined using a bicinchoninic assay protein assay kit (ThermoFischer Scientific, Waltham, MA, USA). A total of 20 µg of proteins were loaded per well on 8% SDS-PAGE gel and migration was performed in migration buffer (Tris-EDTA, SDS, glycine) under 120 V for 90 min. Proteins were transferred onto nitrocellulose blotting membranes (GE Healthcare, Chicago, IL, USA) with a constant amperage of 350 mA for 90 min in transfer buffer (Tris-base, glycine, SDS, methanol). Membranes were blocked for 1 h at room temperature with Tris-buffered saline (TBS) containing 0.1% Tween and 5% BSA. Primary antibodies—decorin (Abcam, ab137508, 1/500), tenomodulin (Abcam, ab203676, 1/500) and collagen III (Abcam, ab7778, 1/5000)—were then incubated on membranes at 4 °C overnight. Membranes were washed in TBS-Tween and incubated with the peroxidase-conjugated secondary antibody for 1 h at room temperature. Finally, membranes were developed on CL-X Posure^®^ films (ThermoFischer Scientific, Waltham, MA, USA) using ECL^TM^ Western Blotting Detection Reagents (GE Healthcare, Chicago, IL, USA). Protein levels were normalized to beta-actin (Sigma, Burlington, MA, USA, A5441, 1/50,000). Films were scanned and quantified by densitometry using ImageJ software (Version 1.52A).

### 4.7. Statistics

Data are presented as means ± SEM. Statistical analysis was performed using an unpaired Student’s *t*-test or the non-parametric Mann–Whitney test. Differences were considered significant at *p* < 0.05 (Prism 5.0, GraphPad Software, Inc., San Diego, CA, USA).

## Figures and Tables

**Figure 1 ijms-23-09928-f001:**
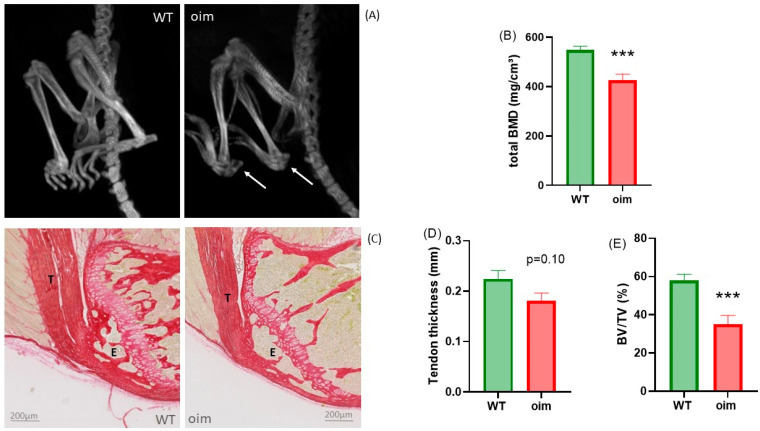
(**A**) Lower limb imaging with in vivo CT of WT and oim mice. Both oim calcanei show avulsion fractures (arrow). (**B**) pQCT total bone mineral density (BMD, mg/cm³) in ulna epiphysis. (**C**) Sagittal section through ulna epiphysis with insertion of triceps brachii tendon on the olecranon of the ulna in WT and oim mice; Sirius Red staining, T = tendon and E = epiphysis of olecranon (ulna). (**D**) Tendon thickness (mm) of triceps brachii tendon. (**E**) Bone volume/total volume (BV/TV, %) ratio in ulna epiphysis, *** *p* < 0.001.

**Figure 2 ijms-23-09928-f002:**
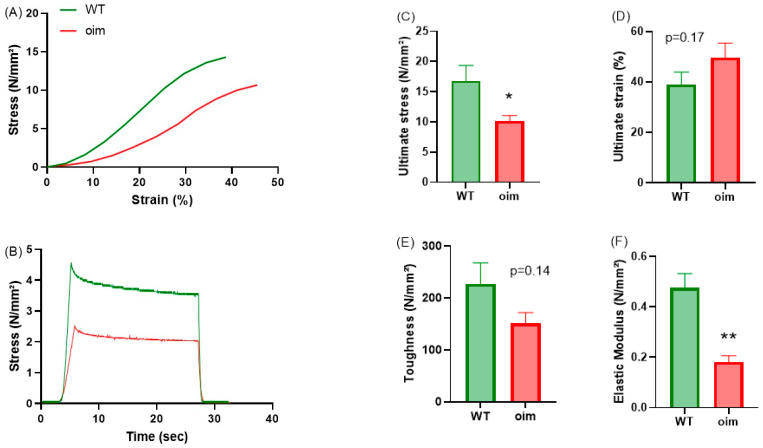
Biomechanical parameters of flexor digitorum longus (FDL) tendons in 5-week-old WT and oim mice. (**A**) Stress–strain curve of one WT mouse (green line) and one oim (red line). (**B**) Evolution of stress through time during a cycle of deformation in the same tendons as in A. (**C**–**F**) Ultimate stress (N/mm^2^), ultimate strain (%), toughness (N/mm^2^) and elastic modulus (N/mm^2^) of FDL tendons. *n* = 7 for oim and 9 for WT mice. * *p* < 0.05, ** *p* < 0.01.

**Figure 3 ijms-23-09928-f003:**
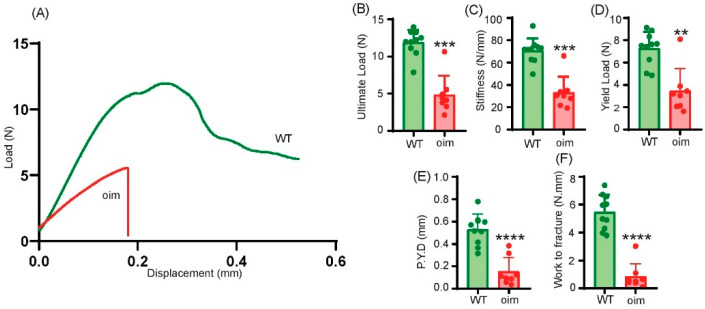
Biomechanical parameters of bone in 5-week-old WT and oim mice. (**A**) Example of load–deformation curves obtained in WT mice (green line) and oim (red line). (**B**) Ultimate load (N), (**C**) stiffness (N/mm), (**D**) yield load (N), (**E**) post-yield displacement (P.Y.D, mm) and (**F**) work to fracture (N.mm) computed from load–displacement curves. *n* = 10 and *n* = 8 for WT mice and oim, respectively. ** *p* < 0.01, *** *p* < 0.001, **** *p* < 0.0001.

**Figure 4 ijms-23-09928-f004:**
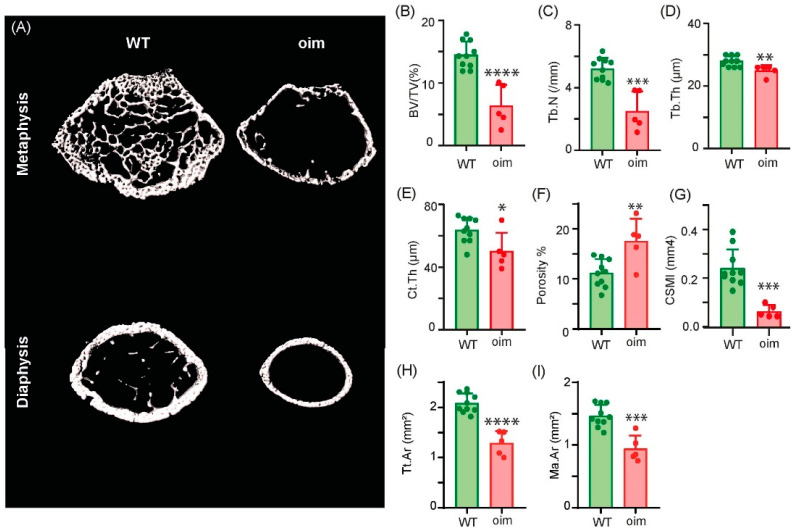
Three-dimensional (3D) models and micro-architecture analysis of trabecular and cortical bone in oim and WT femurs. (**A**) Three-dimensional models of bone metaphysis and diaphysis. (**B**–**D**) Trabecular and (**E**–**I**) cortical microarchitectural parameters. BV/TV = bone volume/total volume, Tb.N = trabecular number, Tb.Th = trabecular thickness, Ct.Th = cortical thickness, CSMI = cross-sectional moment of inertia, Tt.Ar = total area, Ma.Ar = marrow area. *n* = 10 and *n* = 5 for WT mice and oim, respectively. * *p* < 0.05, ** *p* < 0.01, *** *p* < 0.001, **** *p* < 0.0001.

**Figure 5 ijms-23-09928-f005:**
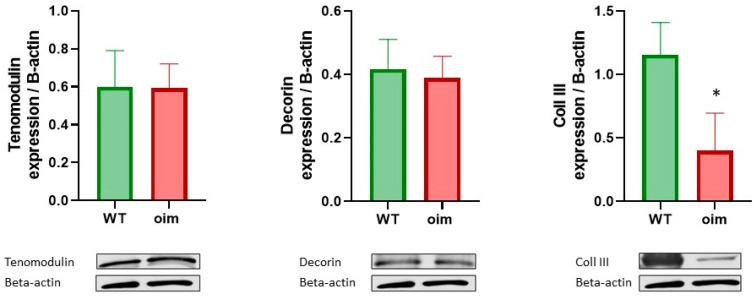
Western blot in tail tendon; the expressions of tenomodulin, decorin and collagen type III are normalized to beta-actin; *n* = 6 for both groups; * *p* < 0.05.

**Figure 6 ijms-23-09928-f006:**
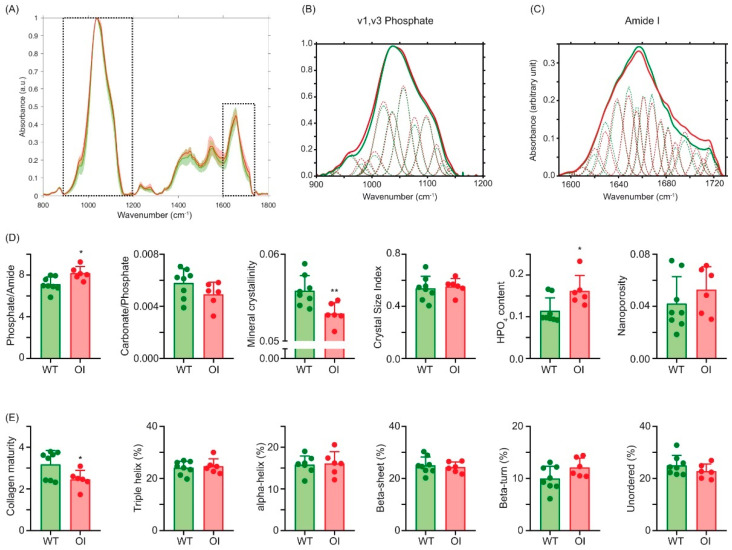
Bone tissue material properties at the femur. (**A**) Mean spectrum in WT mice (green line) and oim (red line). (**B**) Inset of the spectrum in the v1, v3 PO4 band with position, width and intensity of subpeaks determined by second-derivative and curve-fitting spectroscopy. (**C**) Inset of the spectrum in the amide I band with position, width and intensity of subpeaks determined by second-derivative and curve-fitting spectroscopy. (**D**) Mineral and (**E**) organic bone tissue material properties. *n* = 10 and *n* = 6 for WT mice and oim, respectively *: *p* < 0.05; **: *p* < 0.01.

**Table 1 ijms-23-09928-t001:** Tissue material properties of triceps brachii tendon assessed with Raman spectroscopy.

Raman Spectroscopy		WT	Oim	*p*-Value
Spectrum: 800–1800 cm^−1^	Enz. Coll Crosslink IR 1670/1690	1.36 ± 0.10	1.60 ± 0.17	NS
	Pentosidine (AGE) IR 1345/920	2.00 ± 0.10	2.39 ± 0.07	<0.01
	Second conf. of Coll (beta sheet) AR 1610/Amide I	0.07 ± 0.01	0.10 ± 0.01	<0.05
	Hydroxyproline IR 872/920	0.91 ± 0.03	0.85 ± 0.14	NS
	GAG IR 1380/920	0.08 ± 0.02	0.06 ± 0.02	NS
	Porosity (unbound water) IR 1380/920	8.85 ± 0.84	12.53 ± 1.57	0.06
Spectrum: 2700–3800 cm^−1^	Organic matrix-related water IR 3243/2940	0.040 ± 0.006	0.026 ± 0.005	0.09
	NH groups in collagen besides matrix-related water IR 3330/2940	0.035 ± 0.004	0.022 ± 0.003	<0.05
	OH of hydroxyproline besides collagen-related water IR 3457/2940	0.0015 ± 0.003	0.011 ± 0.003	NS

Enz. = enzymatic, Coll = collagen, AGE = advanced glycation end-products, conf. = conformation, GAG = glycosaminoglycan. *n* = 8 for oim and 6 for WT mice, NS = not significant.

## Data Availability

Not applicable.

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
