# Peer review of "Biomechanical, Microstructural and Material Properties of Tendon and Bone in the Young Oim Mice Model of Osteogenesis Imperfecta"

_ijms, 2022, doi:10.3390/ijms23179928_

Round 1

Reviewer 1 Report

Dear Authors

the paper is very interesting and can be considered for publication. However, some topics need to be addressed before acceptance

1) Please discuss potential oral implication for patients affected by osteogenesis imperfecta. Cite DOI 10.1177/1721727X1201000208

2) Please specify the role of stem cells in osteogenesis imperfecta and potential implication in dental field. Please cite PubMed ID33386051 and PubMed ID32811413

Author Response

1) We thank the reviewer for this comment. We have added information about dentinogenesis imperfecta in the introduction:

“Aside from skeletal alterations and symptoms, teeth are also mineralized tissues rich in collagen type I and commonly implicated in OI (4). They exhibit discoloration and translucency and can be worn off prematurely. The roots are short, constricted and the dentine is hypertrophic (5). These changes are characteristic of dentinogenesis imperfecta and can be observed either in the context of OI or isolated. Moreover, patients could also present hearing loss, blue sclera, joint hypermobility, muscle weakness or cardiovascular abnormalities. All these manifestations are listed in supplementary Table, according to other authors (5,6). By extension, all tissues with a high collagen I content, like the tendons, could be affected. Tendon ruptures have been described in patients with OI (6-9). They were sometimes bilateral and occurred at the site of bone insertion, namely the enthesis, which distributes the mechanical forces on the whole tendon-bone interface (10)”.

However, the article DOI 10.1177/1721727X1201000208 deals with tooth orthodontic movement after maxillofacial surgery and does not mention osteogenesis imperfecta. As such, we do not feel appropriate to include the suggested reference in our paper.

2) We thank the reviewer for this comment. We have added information about treatment in the discussion:

“Current treatments for OI aim to increase bone strength, prevent fracture and improve the quality of life. They include a multidisciplinary approach with drug treatment, orthopedic surgery, physiotherapy and preventive dental care (27). Intravenous injections of bisphosphonates are the most widely used intervention to reduce bone resorption but their effects are still debated (26). Novel therapies are under evaluation, including combination of antiresorptive and anabolic drugs or stem cells therapy, which show potential positive effects (27, 28). It would be interesting to know if the treatments used have also an indirect effect on tendons. Indeed, with a stronger bone, we could expect greater use through movement and therefore a possible improvement of tendon properties.”

Reviewer 2 Report

Dear Sirs,

thank you for the opportunity to review this nicely written paper. I would like to add some suggestions of mine:

1. Please present the information given in lines 37-43 in the table or a graph

2. In the introduction, it would be valid to mention that osteogeesis imperfecta is also often combined with dentinogenesis or amelogenesis imperfecta and it would be valid to add some information on that topic

3. In the materials and methods, please add the information if you have a speciffic bioethical commitee permition (please, present the number). If it is not neccesarry, provide the regulation

4. in line 174-175 please explain what you mean by "tissue age"

5. In the discussion, it would be valid to add some information regarding the methods of treatment and dealing with bone and tendon "problems".

After those corrections, the article could be accepted for publication.

Best regards

Author Response

1) We thank the reviewer for this comment. We have added a table summarizing this information.

2) We thank the reviewer for this comment. We have added information about dentinogenesis imperfecta in the introduction:       
“Aside from skeletal alterations and symptoms, teeth are also mineralized tissues rich in collagen type I and commonly implicated in OI (4). They exhibit discoloration and translucency and can be worn off prematurely. The roots are short, constricted and the dentine is hypertrophic (5). These changes are characteristic of dentinogenesis imperfecta and can be observed either in the context of OI or isolated. Moreover, patients could also present hearing loss, blue sclera, joint hypermobility, muscle weakness or cardiovascular abnormalities. All these manifestations are listed in supplementary Table, according to other authors (5,6). By extension, all tissues with a high collagen I content, like the tendons, could be affected. Tendon ruptures have been described in patients with OI (6-9). They were sometimes bilateral and occurred at the site of bone insertion, namely the enthesis, which distributes the mechanical forces on the whole tendon-bone interface (10)”.

Concerning amelogenesis imperfecta, it is generally due to different genetic disorders than those concerning collagen type I matrix.

3) Our experimental procedures on mice were approved by the ethics committee for animal research of the Université Catholique de Louvain. The reference number is 2020/UCL/MD/02 (section Animal Procedures).

4) We thank the reviewer for this comment.           
Tissue age corresponds to the time since first mineralization of the bone tissue. This term is well accepted in the field of bone quality (see reference Pubmed PMID 35167989, 33582361, 32927103). In fact, all parameters assessed for Bone ECM material properties change with ageing of the tissue, that could be seen also as a maturation process of the bone ECM. As an example, the mineralization kinetic follows a bimodal pattern. It is crucial, in order to compare in between group of animals, that zones of same tissue age are recorded. The use of calcein or any calcium fluorescent tracker represents a good approach for this purpose. We hope that this clarifies the term tissue age.

5) We thank the reviewer for this comment. We have added a paragraph regarding the methods of treatment and dealing with bone and tendon in the discussion:          
“Current treatments for OI aim to increase bone strength, prevent fracture and improve the quality of life. They include a multidisciplinary approach with drug treatment, orthopedic surgery, physiotherapy and preventive dental care (27). Intravenous injections of bisphosphonates are the most widely used intervention to reduce bone resorption but their effects are still debated (26). Novel therapies are under evaluation, including combination of antiresorptive and anabolic drugs or stem cells therapy, which show potential positive effects (27, 28). It would be interesting to know if the treatments used have also an indirect effect on tendons. Indeed, with a stronger bone, we could expect greater use through movement and therefore a possible improvement of tendon properties.”